# Course of post COVID-19 disease symptoms over time in the ComPaRe long COVID prospective e-cohort

Viet-Thi Tran [1,2✉], Raphaël Porcher [1,2], Isabelle Pane[1] & Philippe Ravaud[1,2,3]

About 10% of people infected by severe acute respiratory syndrome coronavirus 2 experience post COVID-19 disease. We analysed data from 968 adult patients (5350 person-months) with a confirmed infection enroled in the ComPaRe long COVID cohort, a disease prevalent prospective e-cohort of such patients in France. Day-by-day prevalence of post COVID-19 symptoms was determined from patients' responses to the Long COVID Symptom Tool, a validated self-reported questionnaire assessing 53 symptoms. Among patients symptomatic after 2 months, 85% still reported symptoms one year after their symptom onset. Evolution of symptoms showed a decreasing prevalence over time for 27/53 symptoms (e.g., loss of taste/smell); a stable prevalence over time for 18/53 symptoms (e.g., dyspnoea), and an increasing prevalence over time for 8/53 symptoms (e.g., paraesthesia). The disease impact on patients' lives began increasing 6 months after onset. Our results are of importance to understand the natural history of post COVID-19 disease.

[1] Université Paris Cité, CRESS, INSERM, INRAE, F-75004 Paris, France. [2] Centre d'Epidémiologie Clinique, Hôpital Hôtel-Dieu, AP-HP, 75004 Paris, France. [3] Department of Epidemiology, Columbia University Mailman School of Public Health, 22 W168th St, New York, NY, USA. ✉email: thi.tran-viet@aphp.fr

As of March 2022, about 437 million people worldwide had been infected by the severe acute respiratory syndrome coronavirus 2 (SARS-CoV-2), the pathogen responsible for coronavirus disease 2019 (COVID-19)[1,2]. According to the United Kingdom Office for National Statistics, about 10% of them will experience post COVID-19 disease or "long COVID", that is, the persistence of symptoms such as fatigue, dyspnoea, chest pain, cognitive disturbances, or arthralgia, for several weeks to months after their initial SARS-CoV-2 infection[2,3]. Research has mainly focused on the occurrence of specific long-term complications among hospitalised and non-hospitalised patients recruited during their acute COVID-19 infection[4-11]. To our knowledge, only a handful of studies have investigated the longitudinal evolution of symptoms among patients with persisting symptoms (i.e., with post COVID-19), but they have generally been small, retrospective or limited to single centres[12-14]. In this study, we used data from a large nationwide cohort of patients to reconstruct the day-by-day course of their symptoms from onset to 1 year after the acute phase of the infection.

## Results

**Participants.** The ComPaRe long COVID cohort is an ongoing nationwide e-cohort of patients with post COVID-19 disease in France, nested in the ComPaRe research programme (www.compare.aphp.fr), an umbrella e-cohort of patients with chronic conditions[15]. The cohort started in December 2020, and recruitment is ongoing. Among the 1859 patients included in the ComPaRe long COVID cohort on October 15, 2021, we analysed the data from the 968 patients reporting: (1) a laboratory-confirmed COVID-19 infection with a positive test for SARS-CoV2 by PCR swab and/or serologic assay; (2) symptoms persisting for at least 2 months after their onset; and (3) enrolment in the cohort by August 1, 2021, and thus at least 2 months of follow-up in this analysis (Fig. 1). To improve the representativeness of results, we weighted observations by calibration on margins, so that the weighted distribution of age (<24, 25–34, 35–49, 50–69, and ≥70 years), gender, and hospitalisation during

the acute phase of the disease (Yes/No) matched the data from the UK Office of National Statistics Covid Infection Survey (2 September 2021 data)[16,17]. Raw and weighted patients' characteristics are presented in the Table 1. Hereafter, results presented always come from the weighted dataset. Raw results are available in the Supplementary Materials.

In the weighted data, patients' median age was 48 years (interquartile range 32–56) with 57.7% (559/968) men. In all, 35.1% (340/968) reported comorbidities, 6.3% (61/968) had chronic lung diseases, and 4.2% (41/968) high blood pressure. The median time between symptom onset and the last follow-up was 174 days (IQR 97 to 284 days). Among participants, 75 (7.7%) had been hospitalised during their acute disease and 34 (3.5%) had been admitted to an intensive care unit (ICU).

**Probability of symptom persistence at 12 months.** Participants were followed up every 60 days with online questionnaires available on computer or smartphone. At each observation point, patients were first asked if they still had symptoms related to COVID-19. Those reporting the persistence of symptoms

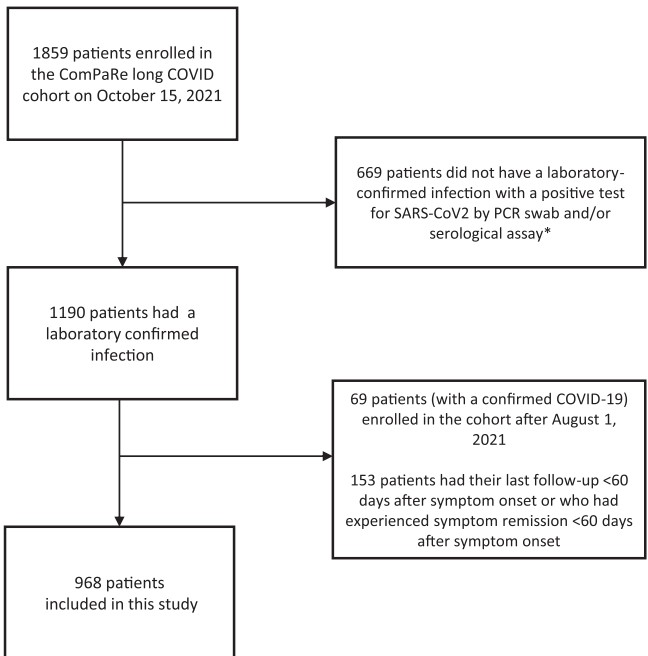

**Fig. 1 Study flow chart.** *The high number of patients without a confirmed infection is due to the limited availability of testing during the first wave of COVID-19 in March 2020, in France.

### Table 1 Patients' characteristics (n = 968).

| Characteristic | Raw data (n = 968) | Weighted data (n = 968) |
|---|---|---|
| Age, median (Q1–Q3)—year | 47 (38–54) | 48 (32–56) |
| Age categories—number (%) | | |
| <24 | 25 (2.6) | 147 (15.2) |
| 24–34 | 136 (14) | 138 (14.3) |
| 35–49 | 448 (46.3) | 259 (26.8) |
| 50–69 | 337 (34.8) | 335 (34.6) |
| >70 | 22 (2.3) | 89 (9.2) |
| Male sex—number (%) | 201 (20.8) | 559 (57.7) |
| Educational level—number (%) | | |
| Middle school or equivalent | 75 (7.7) | 70 (7.2) |
| High school or equivalent | 105 (10.8) | 147 (15.2) |
| 2 years post-secondary education | 216 (22.3) | 178 (18.4) |
| ≥3 years post-secondary education | 553 (57.1) | 559 (57.8) |
| Other | 19 (2.0) | 14 (1.5) |
| At least one comorbidity—number (%) | 382 (39.5) | 340 (35.1) |
| Comorbidities—number (%) | | |
| High blood pressure | 45 (4.6) | 41 (4.2) |
| Diabetes | 23 (2.4) | 25 (2.6) |
| Stroke or cardiac ischaemic disease | 5 (0.5) | 6 (0.6) |
| Chronic kidney disease | 2 (0.2) | 1 (0.1) |
| Chronic lung disease (e.g., asthma/COPD) | 71 (7.3) | 61 (6.3) |
| Thyroid disorder | 25 (2.6) | 17 (1.7) |
| Cancer | 18 (1.9) | 18 (1.9) |
| Depression/Anxiety | 42 (4.3) | 40 (4.1) |
| Time since symptom onset, median (Q1–Q3)—days | 192 (97–297) | 174 (97–284) |
| Hospitalised for COVID-19—number (%) | 156 (16.1) | 75 (7.7) |
| Hospitalised in ICU for COVID-19—number (%) | 42 (4.3) | 34 (3.5) |
| Duration of hospitalisation, median (Q1–Q3) | 7 (1–14) | 12 (3–26) |

Weighted data were obtained by calibration on margins with weights for age (<24, 25–34, 35–49, 50–69, and ≥70 years old), gender and hospitalisation during the acute phase of the disease, derived from the data from the Office of National Statistics in the United Kingdom.

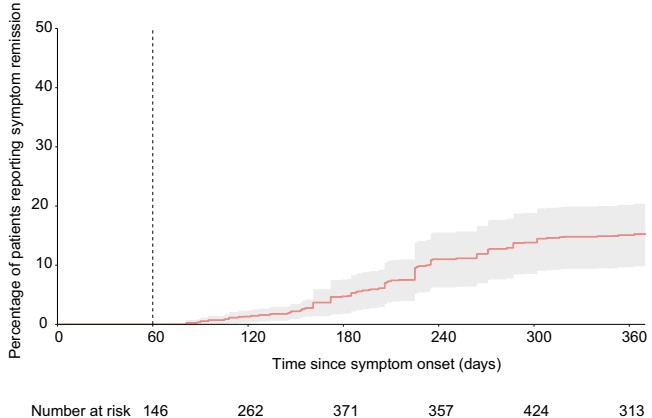

**Fig. 2 Cumulative event curve for remission of post COVID-19 symptoms.** Time of remission was defined as the first time that patients reported no longer experiencing any symptoms of post COVID-19 disease. The time at risk started at entry in the cohort and ended on October 10, 2021. Follow-up data were censored at the participants' latest observation point. Error bands represent 95% confidence intervals. Source data are provided as a Source Data file.

completed the long COVID symptom tool (ST) and impact tool (IT), a pair of validated patient-reported instruments assessing respectively 53 symptoms and 6 dimensions of patients' lives that can be affected by the disease[18]. Those reporting that they no longer had any symptoms were asked to report the date when they first noticed the absence of symptoms. Overall, our data covered 5350 person-months, with a median follow-up since cohort enrolment of 181 days (interquartile range 118 to 240 days). The proportion of patients lost to follow-up was <20% at every follow-up (Supplementary Material 1).

Remission of symptoms (i.e., disappearance of all symptoms) was observed during follow-up for 150 patients. Among those who reported full symptom remission, 50/150 (33.3%) subsequently described a relapse for at least one symptom. At 12 months, the probability of symptom persistence (including patients in remission who relapsed) was 84.9% (95% CI 79.8–90.4%) (Fig. 2).

**Day-to-day prevalence of post COVID-19 disease symptoms.** We used multistate models taking both left and right censoring into account to estimate the longitudinal evolution of symptoms. In the models, each patient contributed to the data during his/her follow-up period. Because our cohort included patients enroled at different times after their initial infection, we were able to reconstruct the prevalence of each symptom, day-by-day, in the study population (Fig. 3A and Supplementary Material 2). First, 27 (51%) symptoms showed a progressive decrease in prevalence over time. Among them, loss of appetite, change/ loss of taste, and cough changed most (>20% decrease). Second, 18 (34%) symptoms showed no specific change in prevalence over time. Among them, word finding problems and dyspnoea were the most prevalent. At 360 days, they affected respectively 48% and 44.5% of the population. Finally, eight symptoms showed an increase in prevalence over time. Among them, neck, back, and low back pain and paraesthesia changed most markedly (>10% increase) (Supplementary Material 3). The evolution of symptoms over time in subgroups by age, sex, and presence of comorbidities is presented in Supplementary Materials 4–6.

Post COVID-19 disease is a relapsing-remitting disease. Sixty days after symptom onset, most of patients reported permanent, daily or weekly symptoms. Over time, relapses became less frequent, with a decrease in the proportion of patients reporting weekly or more frequent symptoms (including those reporting permanent symptoms) and a parallel increase in the proportion reporting relapses less than weekly (Supplementary Material 7).

**Evolution over time of the impact of post COVID-19 disease on patients' lives.** Figure 3B illustrates patients' perceptions of the impact of the disease on their lives, measured by the long COVID IT. In particular, we present the proportion of patients reporting an unacceptable disease state, defined as a score on the long COVID IT above the value at which >75% of patients consider that they could not cope with a similar level of lifelong symptoms[18]. This analysis revealed two distinct phases of the disease. In a first phase, from 60 to 180 days, the burden of disease progressively decreased as several symptoms disappeared, while the number of patients reporting an unacceptable symptom state diminished slowly to 50%. After 6 months, the proportion of patients reporting an unacceptable disease state increased more rapidly, with 60–70% of patients considering their disease unacceptable at that time. This secondary increase may correspond to patients' realisation that they had a chronic disease. This U-shaped trend in the evolution of the perception of the impact of the disease on patients' lives over time was found in subgroups defined by age and sex. (Supplementary Material 8 and 9).

**Discussion**
This study reports how the symptoms and impact of post COVID-19 evolve after the acute phase of the disease, in a large prospective cohort of patients with a laboratory-confirmed infection. Among patients with post COVID-19 disease, 85% still reported symptoms 1 year after symptom onset. This finding is consistent with observations in a single-centre study in Germany, where 20% of patients were free from all 14 symptoms under study at 12 months[12].

The course of symptoms over time highlighted three distinct patterns that offer insight into the aetiologies and mechanisms underlying this disease. First, we observed a decrease in prevalence over time for symptoms such as loss of taste or smell, coughing, or diarrhoea. For example, the prevalence of coughing decreased from 50 to 20% of participants within the first 6 months after symptom onset before reaching a plateau. This was similar for loss of smell, with a plateau reached after 8 months. This evolution, indicating recovery from the acute phase, is slower than expected. Indeed, most guidelines still consider a cut-off of 12 weeks to distinguish ongoing symptomatic COVID-19 (i.e., signs and symptoms of COVID-19 from 4 to 12 weeks) from post COVID-19 syndrome[19]. For other symptoms, we showed that their prevalence increased over time. For example, the prevalence of hair loss increased over time with 8% and 15% of participants reporting it at 2 months and 1 year after onset respectively. Late symptom appearance, especially alopecia, has been reported in other studies and should be further investigated[20]. Finally, symptoms showing no change of prevalence over time may be caused by mechanisms that do not change rapidly over time, such as deconditioning or post-traumatic stress disorder, or due to a mixture of recovery from acute disease and late-onset symptoms appearing as a consequence of COVID-19[21].

We found some differences in the evolution of symptoms between men and women; and in age groups. We hypothesise that differences may be due either to different underlying causes of the persistent symptoms or to specific factors directly affecting symptoms. For example, several studies have highlighted differences between the sexes in immune responses (in terms of levels

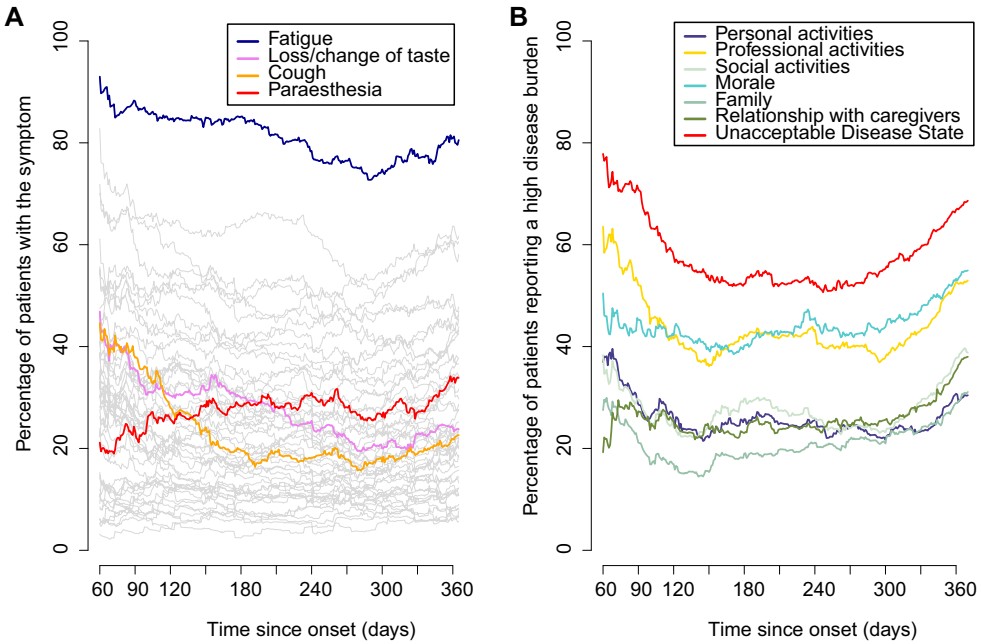

**Fig. 3 Day-by-day trends in the prevalence of post COVID-19 disease symptoms (A) and of their impact on patients' lives (B). A** The figure presents the day-by-day prevalence of each of the 53 symptoms assessed by the Long COVID ST (grey lines). Examples of specific symptoms have been highlighted (coloured lines). For each symptom and at each observation point, we assumed that patients could either be "experiencing" or "not experiencing" the symptom. We assumed that their state at an arbitrary time was the same as the state at their previous observation point and that their states before their first observation and after their last observation are unknown. **B** The figure presents the day-by-day evolution of the six domains of patients' lives that can be affected by post COVID-19 disease and are assessed by the Long COVID IT. For each item and at each observation point, we modelled patients answers as either "reporting" a significant impact of the disease on this domain" (i.e., item score >7) or "not reporting" this impact (i.e., item score <8). We assumed that their state at an arbitrary time was the same as the state at their previous observation point and that their states before their first observation and after their last observation are unknown. The red lines represent a similar model for the Patient Acceptable Symptomatic State (PASS) of the long COVID IT, which is the long COVID IT score below which 75% of patients find that their disease state is acceptable. Source data are provided as a Source Data file.

of innate immune cytokines such as IL-8 and IL-18, induction of non-classical monocytes, and T cell activation)[22] and in response to a traumatic event[23].

Our results demonstrate the substantial impact of post COVID-19 disease on patients' lives. Specifically, the course of patients' perceptions of the impact of the disease changed similarly over time across all subgroups, with an aggravation 6 months after onset. We hypothesise that this corresponds to patients' realisation that this novel and previously unknown disease is chronic rather than acute. This trend seemed more pronounced among younger patients, for whom post COVID-19 is more likely to be both their first contact with a chronic disease and a biographical disruption of their lives[24].

Strengths of this study lie in the prospective follow-up of a large population of patients with a confirmed COVID-19 infection and their regular assessment of their symptoms using validated patient-reported outcome measures, developed from patients' lived experience, with excellent reproducibility (intraclass correlation coefficient 0.83, 95% confidence interval 0.80–0.86)[18]. The use of online questionnaires reduced social desirability bias and may provide better estimates of the prevalence of sensitive symptoms such as genitourinary or cognitive disorders[25].

This study has several limitations. The recruitment of volunteers in the ComPaRe long COVID cohort may have selected patients who had overall more symptoms, and our recruitment strategy included a social media and general media campaign that may have selected younger and better educated patients. Second, our sample included a majority of women. Persistent symptoms after COVID-19 are more frequent among women; for example,

the odds of persistent fatigue at 1 year among women compared with men was 1.43 [1.04–1.96][26]. To minimise these issues, we used a weighted dataset with weights derived from the September 2021 data from the Office of National Statistics in the United Kingdom[16]. Yet, because our original dataset included fewer men, results in this subgroup may lack precision. More generally, the subgroup analyses were post-hoc, and should be considered as exploratory. Third, in this study we included patients with a history of confirmed SARS-CoV-2 infection who had symptoms for at least 2 months, in accordance with the WHO's recent consensus definition of post COVID-19 disease[27]. However, our data prevented us from ascertaining whether an alternative diagnosis might explain the symptoms. Fourth, as patients could enrol in the cohort at any time point after their initial infection (with an interval between symptom onset and enrolment exceeding 300 days for 25% of participants), we chose to minimise memory bias by not asking retrospectively for symptoms experienced during the infection's acute phase. This limits our ability to describe how symptoms change immediately after the acute phase of the disease. Fifth, disease remission was defined as having no symptoms among 53. As the likelihood of having at least one symptom, at any time point, is expected to be high even in the general population, our analyses may overestimate the number of patients who still report symptoms 1 year after their symptom onset. Sixth, this study did not use a control group. Because several symptoms of post COVID-19 disease are non-specific, we cannot ascertain that the observed symptoms are different from or in addition to what might be due to intercurrent illnesses, comorbidities, ageing, or social effects of living through the pandemic. Finally, in view of the limited number of patients

who were hospitalised in ICUs in our study, our results cannot be generalised to this specific population.

In conclusion, our study shows that most patients with post COVID-19 disease have symptoms evolving in different patterns but persisting through 1 year. Recovery from the acute infection is a slow process, and the prevalence for most symptoms decreased over time before plateauing 6–8 months after onset. Our results are of importance to understand the natural history of this disease, and should help physicians to inform their patients about the potential course of this disease.

## Methods

This research complies with all relevant ethical regulations. It was approved by the Institutional Review Board of Hôtel-Dieu Hospital, Paris (IRB: 0008367). All patients provided informed consent electronically before participating in the e-cohort, which was considered equivalent to written, informed consent for the study participants by the ethics and regulatory bodies which authorised the study, in France.

**Data sources and participants**. The ComPaRe Long COVID cohort is an ongoing nationwide e-cohort of patients with post COVID-19 disease in France, nested in the ComPaRe research programme (www.compare.aphp.fr), an umbrella e-cohort of patients with chronic conditions[15]. The cohort began in December 2020 and recruited participants through: (1) a social and general media campaign, (2) calls for participation from partner patient associations and on the official French contact tracing app "TousAntiCOVID", and (3) a "snowball" sampling method where participants were encouraged to invite people who had a COVID-19 infection and persisting symptoms to enrol[28].

Participants interested in participating in ComPaRe can enrol themselves on the cohort's online website (https://compare.aphp.fr). All patients provide electronic consent before participating in the e-cohort. After completing initial demographic and clinical information, patients receive invitations by e-mail to answer online questionnaires related to their diseases and/or treatment, as well as prompts to participate in research projects nested in ComPaRe.

**Participants**. In this study, we analysed the data from patients (1) reporting a laboratory-confirmed COVID-19 infection, with a positive test result for SARS-CoV2 by PCR swab and/or a serological assay; (2) reporting at least one symptom persisting 2 months after symptom onset, in a validated list of 53 symptoms;[18] and (3) who enrolled in the cohort by August 1, 2021, so that they would have at least 2 months of follow-up. The 2-month interval was chosen according to the WHO's definition of post COVID-19 disease[27]. Participants' characteristics are reported in the Table 1.

**Measurement of post COVID-19 disease symptoms and impact**. Participants were followed-up every 60 days with online questionnaires. They received an e-mail invitation prompting them to connect on the ComPaRe (https://compare.aphp.fr) secure internet platform, which is accessible by computer or smartphone. At each observation point, patients were first asked if they still had symptoms related to COVID-19. Those reporting that they still had persisting symptoms completed the long COVID ST and IT, a pair of validated patient-reported instruments assessing respectively 53 symptoms and 6 dimensions of patients' lives that can be affected by the disease[18]. Those reporting that they no longer had any symptoms were asked to specify the date when they first noticed the absence of symptoms.

Patients received reminders every 15 days encouraging them to complete the online questionnaires. Of note, the 53 symptoms of the long COVID ST fully overlap with those from the WHO's "Clinical Platform Case Report Form (CRF) for Post COVID condition (Post COVID-19 CRF)"[29].

Results for the long COVID IT were dichotomised by their Patient Acceptable Symptom State, which is the score below which 75% of patients considered their symptom state acceptable. Results from the specific items of the long COVID IT were dichotomised by their item score (>7 or ≤7).

The relapsing-remitting nature of post COVID-19 was investigated by asking participants to describe how often symptoms occurred, using four response options: "permanent symptoms", "daily relapses", "weekly relapses", and "less than weekly relapses". Those reporting that they no longer had any symptoms were asked to report the date when they first noticed the absence of symptoms.

The vital status of ComPaRe participants is assessed regularly by linkage with the French National Death Register (INSEE, fichiers des personnes décédées). As of October 28, 2021, no patient enroled in the cohort had died.

**Analyses**. Because we used data from an e-cohort, we had no missing data within validated online questionnaires.

To enhance the representativeness of our estimates, we used a weighted dataset obtained by calibration on margins with weights for age (<24, 25–34, 35–49, 50–69, ≥70 years old), gender and hospitalisation during the acute phase of the disease derived from the data from the Office of National Statistics in the United Kingdom[16,17]. Weights were truncated at the 99th percentile.

The Kaplan–Meier method was used to estimate the cumulative probability of remission of all post COVID-19 symptoms. The date of symptom remission was defined as the first date when patients no longer reported any symptoms. A single report of all symptoms resolving at any point during follow-up was enough to be classified as remission. The time at risk started at disease onset and ended on October 10, 2021. Data were left truncated at the time of patients' enrolment in the cohort. Follow-up data were censored at the participants' latest observation point.

The longitudinal evolution of symptoms, their frequency of occurrence and their impact on patients' lives were studied by using multistate models accounting for both left and right censoring. Under the assumption that a patient's state (e.g., "experiencing the symptom" or "not experiencing the symptom") at an arbitrary time was the same as the state at their previous observation time and that patient's states before their first observation and after their last observation are unknown, we estimated the day-by-day prevalence, since disease onset, of the state of interest in our population. Therefore, each patient contributed to the estimation of the prevalence of the state of interest during the time of his/her follow-up.

We performed several post-hoc subgroup analyses of the evolution of symptoms and impact on patients' lives by age (≤40 and > 40 years), sex and presence of at least one comorbidity (yes/no).

Statistical analyses were performed with R software (http://www.R-project.org, the R Foundation for Statistical Computing, Vienna, Austria), version 4.0.5 and the *msm* package[30].

**Reporting summary**. Further information on research design is available in the Nature Research Reporting Summary linked to this article.

## Data availability

All data generated and used in the study are from the ComPaRe e-cohort platform. All data presented in this study are available to public research teams for health-related research in the public interest. Researchers who wish to access the data must comply with the conditions detailed on https://compare.aphp.fr. Contact for access requests is contact.compare@aphp.fr. All requests for data are examined by the ComPaRe scientific committee and responded within 3 months from submission. The processed data from Figs. 2 and 3 are provided in the Source Data file. Source data are provided with this paper.

## Code availability

All statistical codes required to reanalyse the data are available upon request.

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

## Acknowledgements
The authors thank Elise Diard, Carolina Riveros and Clara Marre for their work in the ComPaRe e-cohort and Jo Ann Cahn for editing.

## Author contributions
Generated the idea: V.T.T.; Conceived and designed the experiments: V.T.T., R.P. and P.R.; Collected data: V.T.T., I.P. and P.R.; Analysed data: V.T.T. and RP; Wrote the first draft of the manuscript: V.T.T.; Contributed to the writing of the manuscript: V.T.T., R.P., I.P. and P.R.; ICMJE criteria for authorship read and met: V.T.T., I.P., R.P. and P.R.; Agree with manuscript results and conclusions: V.T.T., R.P., I.P. and P.R. V.T.T. is the guarantor, had full access to the data in the study, and takes responsibility for the integrity of the data and the accuracy of the data analysis.

## Competing interests
The authors declare no competing interests.
