## [Peer Review File · Nature Communications]

Reviewers' comments:

Reviewer #1 (Remarks to the Author):

In this study, patients with persistent symptoms of different types after an acute phase of COVID infection were followed longitudinally between 2 and 7 months using an app. It is a fairly good representation of the affected population, as about 15% are hospitalised and the rest are outpatients. In addition, it is a multicentre study, which is rare in this field. It is important to describe the symptoms in the course to understand if there is a remission of symptoms.

I see major difficulties in the design of the study, methods, data analysis and finally the scientific deductions made from these observations.

- 1) One major difficulty I see is the completely inhomogeneous groups that were included. The time period of a past COVID ascertainment diverged between 2 and 6 months.
- 2) How long is COVID defined?
- 3) As colleagues describe very well, symptoms change in the first 240 days after the acute phase of infection. This shows that a precise definition of long COVID is essential to include people. The question of this paper should therefore be: how do symptoms change after the acute phase of the disease? This is a very s'important question, but it is not correctly presented in the current design and research question.
- 4) The drop-out rate is relatively high. Therefore, the longitudinal observations can ultimately only be based on. It would make more sense and be cleaner to include only those who were in the study for at least 6 months.
- 5) The presentation of the data is very superficial. Nothing is known about the clinical condition during the acute phase of infection.
- 6) Since it was via an app, only a relatively young group that is tech-savvy. This possibly represents a bias.
- 7) It is not a day-by-day assesmnet as the pat. were surveyed every 60 days via an app.
- 8) There was not a single face-to-face contact. It is a weakness that there are only online questionnaires without clinical assesment.
- 9) Finally, these observations do not suggest potential pathophysiological mechanisms underlying long COVID.

Reviewer #2 (Remarks to the Author):

This is a longitudinal study of patients with persistent symptoms after COVID-19 in France. The authors have followed patients with confirmed SARS-CoV-2 infection every second month for up to one year, minimum 2 months. The study was online, meaning that participants were recruited by social media and patient associations. Recruitment required participants to having had symptoms for 3 weeks after diagnosis.

The recruitment procedure is the main weakness of the study. Having recruited 837 participants

nationally, out of France's 6,8 million confirmed cases, makes risk of selection bias obvious. This is further confirmed by having achieved 80% women among the participants, while approx 50% of the infected are male. Statements in the abstract such as 88% having symptoms after 9 months can easily be interpreted as a prevalence, even though not specifically stated as such, and should be removed. The interesting observations in the study are the observations of symptom changes over time. Again, a more rigid recruitment may have led to other changes in symptoms, but the observations are likely representing true longitudinal symptom patterns. The differences between younger and older participants is also of interest.

Specific comments:

1. Are all participants recruited 3 weeks after disease onset, or is it possible that some are recruited much later? This will affect the validity of the longitudinal data.
2. What was the reasoning behind requiring 3 weeks duration of symptoms?
3. 80% of participants are women, and women have more symptoms of fatigue in general. Was any calculations performed on gender differences?
4. Figure 1 shows cumulative remission of symptoms. If a substantial portion of participants were recruited much later than 3 weeks, there is a risk of over-representation of long-haulers. I would like to see an analysis of early and late recruited participants, if not all were recruited at 3 weeks (see my point 1 above).
5. Figures 2 and suppl 5 are interesting. How much is the data affected by gender, and again, by recruitment time. Particularly the upwards trend from 6 to 9 months in fig 2B needs to be explained by more analysis regarding recruitment time after infection. Are there age and gender differences in the trends in this panel?
6. Were any calculations made on the effect of comorbidity? I would like to see more statistical calculation on how symptom scores and duration were affected by comorbidity, age, gender, and other factors recorded as background variables.

Reviewer #3 (Remarks to the Author):

Reviewer: Shamil Haroon

General comments

This paper describes a prospective cohort study on individuals with Long COVID who were SARS CoV-2 RT-PCR positive or had indicative serology. The study includes important and novel findings relevant to understand the natural history of Long COVID. The strengths of the study include the wide range of symptoms captured, the longitudinal data capture, as well as the inclusion of data on the impact of symptoms on quality of life. A key limitation is the source population that was recruited by social media and not randomly sampled from a pre-specified sampling frame. There is also no control population to compare the difference in symptoms, making it difficult to infer the extent to which the pattern of symptoms observed can be attributed to Long COVID as opposed to other factors such as comorbidities or social and environmental factors affecting the study population. A number of key reporting items were missing from the manuscript and it was sometimes difficult to assess because much of the key

material is in the supplementary file. I would recommend writing the report in accordance with the STROBE statement for cohort studies as far as possible, although I appreciate that there are limits set by the journal format that may not allow this to be wholly possible within the main text.

Specific comments

Abstract

- The abstract does provide sufficient details on the study population. It should clarify if these are hospitalised or non-hospitalised patients, which setting they were recruited from (population-based, primary care, secondary care, critical care), whether they were adults or children, etc.
- The abstract should include a brief description of the Long COVID Symptom Tool. It is unclear whether it is a paper-based questionnaire, an online questionnaire or an app-based platform.
- The abstract reports the change in prevalence of symptoms over time but does not consistently state over what time period.

Background

- The first paragraph of the background section mentions that long COVID is the persistence of symptoms after initial SARS CoV-2 symptoms. I think this should be reworded to persistence of symptoms after initial SARS CoV-2 infection.
- Note that the UK study referred to uses the term “post-covid-19 syndrome” rather than “post-acute COVID” to denote symptoms lasting beyond 12 weeks.
- The background section mentions that only a handful of studies have investigated the longitudinal evolution of symptoms with Long COVID and that they are either small, retrospective or limited to single centres and cites two papers. The study by Davis et al that has been cited actually has quite a large sample size (n=3762) from 56 countries. Other relevant studies include Blomberg et al 2021 (Long COVID in a prospective cohort of home-isolated patients), and Sudre et al 2021 (Attributes and predictors of Long COVID).

Results

- It would be helpful to briefly mention whether the patients were recruited from a population-based cohort, from primary care, or secondary care, and the method of recruitment.
- When did study recruitment commence?
- Point 2 of the eligibility criteria states that participants should have had symptoms persisting three weeks after onset but doesn't clarify what is meant by “onset”. Is this onset of symptoms or test result?
- Further to the above point, what symptoms were included as part of the eligibility criteria and how were these symptoms selected?
- Did any participants die during follow-up? How was participant death dealt with in the analysis? Was follow-up censored at this point and how was this handled in the analysis?
- What was the level of data completeness and how was missing data handled in the analyses? How complete was follow-up?
- How was remission of symptoms classified? Was a single report of all symptoms resolving at any point during follow-up classed as remission or did the symptoms need to have resolved for a particular length of time?
- The probability of symptom persistence is reported at 6 and 9 months as 95.6% and 88.7%. Is this referring to those with a relapse of symptoms?

- Were subgroup analyses performed to assess whether the observed trends varied by specific characteristics such as age, sex, or hospitalisation status? The results seem to suggest this was done by age but there is no specific mention of this in the methods.

Figures

- There appears to be an error in the y-axis title for figure 1. It indicates % of patients with remission of symptoms. However, this is actually the proportion rather than the percentage.
- The opposite is true for the y-axis title for figure 2a. This states proportion of patients where it is the percentage.
- The similarity of some of the colours in figure 2b make it difficult to interpret.

Discussion

- The authors suggest that the timeframes in NICE definitions of Long COVID should be revised. However, the rationale for this argument could be more clearly explained.
- The explanation for a rise in prevalence of certain symptoms such as memory problems was also unclear. This could be phrased more clearly.
- It would be helpful to guideline developers to include a clear statement on the implications of the findings for the management of Long COVID going forward.

Other

- The sections on acknowledgements and contributions do not appear to have been completed.

Please find below a point-by-point answer to all reviewers' comments for our manuscript titled "*Course of post COVID-19 disease symptoms over time in the ComPaRe long COVID prospective e-cohort*" (NCOMMS-21-29487-T).

Reviewer #1 (Remarks to the Author):

In this study, patients with persistent symptoms of different types after an acute phase of COVID infection were followed longitudinally between 2 and 7 months using an app. It is a fairly good representation of the affected population, as about 15% are hospitalised and the rest are outpatients. In addition, it is a multicentre study, which is rare in this field. It is important to describe the symptoms in the course to understand if there is a remission of symptoms.

I see major difficulties in the design of the study, methods, data analysis and finally the scientific deductions made from these observations.

1) One major difficulty I see is the completely inhomogeneous groups that were included. The time period of a past COVID ascertainment diverged between 2 and 6 months.

Answer: Patients can enrol in the ComPaRe long COVID cohort at any time after their COVID infection.

In this study, we used multistate models that allow us to take both left and right censoring into account to reconstruct the evolution of patients' symptoms since disease onset. With such methods, we can reconstruct the evolution (at the population level) of each symptom since the onset of the disease. At each discrete time point (e.g., 90 days after onset), the prevalence of the symptom in the population is based on the data from patients who were actually followed at that time (e.g., for patients who had an observation before and after 90 days after onset). Therefore, each patient contributed to the estimation of the prevalence of symptoms over time during his/her follow-up time (from 2 months to 10 months, depending on their date of enrolment in the cohort). In fact, we were able to perform this analysis because we had a large number of patients who enrolled in the cohort at heterogeneous times after symptom onset.

We have modified the text and the online methods to underline our use of methods that take both left and right censoring into account.

We used multistate models taking both left and right censoring into account to estimate the longitudinal evolution of symptoms; each patient contributed to the data during his/her

follow-up period. Because our cohort included patients enrolled at different times after their initial infection, we were able to reconstruct the prevalence of each symptom, day by day, in the study population

In the online methods:

The longitudinal evolution of symptoms, their frequency of occurrence and their impact on patients' lives were studied by using multistate models accounting for both left and right censoring. Under the assumption that a patient's state (e.g., "experiencing the symptom" or "not experiencing the symptom") at an arbitrary time was the same as the state at their previous observation time and that patient's states before their first observation and after their last observation are unknown, we estimated the day-by-day prevalence, since disease onset, of the state of interest in our population. Therefore, each patient contributed to the estimation of the prevalence of the state of interest during the time of his/her follow-up.

2) How long is COVID defined?

Answer: In the original paper, we defined long COVID as having: 1) a laboratory confirmed SARS-CoV2 infection and 2) at least one symptom among the 53 evaluated by the Long COVID Symptom Tool persisting more than 3 weeks after the initial infection (these 53 symptoms fully overlap with those from the WHO Long COVID case form). The delay of 3 weeks was chosen in October 2020 because, at this time, preliminary data on persisting symptoms after a SARS-CoV2 infection suggested this threshold to define long COVID (Tenforde, Morbidity Mortality Weekly Report, 2020).

In the revised paper, we have reanalysed our data in light of the recent definition of post COVID-19 disease issued by the WHO on the October 6, 2021, and restricted our analyses to patients with a confirmed infection and for whom we could ascertain that they had had at least one symptom for >60 days.

We have clarified the inclusion criteria in the Online Methods

In this study, we analysed the data from patients 1) reporting a laboratory-confirmed COVID-19 infection, with a positive test result for SARS-CoV2 by PCR swab and/or a serological assay; 2) reporting at least one symptom persisting two months after symptom onset, in a validated list of 53 symptoms; and 3) who enrolled in the cohort by August 1, 2021, so that they would have at least 2 months of follow-up. The two-month interval was chosen according to the WHO's definition of post COVID-19 disease.

We also underline in the discussion that our definition is limited because we could not ascertain that symptoms experienced by patients could not be explained by an alternative diagnosis.

Third, in this study we included patients with a history of confirmed SARS-CoV-2 infection who had symptoms for at least two months, in accordance with the WHO's recent consensus definition of post COVID-19 disease. However, our data prevented us from ascertaining whether an alternative diagnosis might explain the symptoms.

3) As colleagues describe very well, symptoms change in the first 240 days after the acute phase of infection. This shows that a precise definition of long COVID is essential to include people. The question of this paper should therefore be: how do symptoms change after the acute phase of the disease? This is a very important question, but it is not correctly presented in the current design and research question.

Answer: We thank the reviewer for the comment. We now emphasize this research question in the introduction and discussion sections of the paper.

In this study, we used data from a large nationwide cohort of patients to reconstruct the day-by-day course of their symptoms from onset to 1 year after the acute phase of the infection

And, in the discussion section:

This is the first study to report how the symptoms and impact of post COVID-19 evolve after the acute phase of the disease, in a large prospective cohort of patients with a laboratory confirmed infection.

4) The drop-out rate is relatively high. Therefore, the longitudinal observations can ultimately only be based on. It would make more sense and be cleaner to include only those who were in the study for at least 6 months.

Answer: We respectfully disagree with the reviewer, but recognize that the confusion may have arisen from how data were presented (from 65% to 80% available data in the original manuscript). Missing data at each time point was <20%, which is considered acceptable in Epidemiology textbooks (Sackett DL, Evidence based medicine: how to practice and teach EBM, 1997).

We understand the confusion as we displayed in the original supplementary material only the number and proportion of “available” data. Unavailable data may be due to 1) patients who have not yet reached the time point; 2) patients who have reached the time point <30 days ago and may still answer the online questionnaire; and 3) patients actually lost to follow-up. This is important because only the latter reason is likely to lead to informative censoring and bias. For other patients, we can assume that their survival prospects are similar to those of participants who continue to be followed and that censoring was not informative.

We have modified the text to highlight the low number of patients actually lost to follow-up:

In all, the proportion of patients lost to follow-up was <20% at all observation points

We completed the supplementary appendix with the number of patients lost to follow-up. For example, 17.4% have a follow-up in the cohort > 3 months but did not answer the online questionnaire sent two months after their enrolment and their data are considered missing. Similar, all patients included in the analysis can still answer the online questionnaire for the 10-month observation after their enrolment in the cohort, and therefore we consider that there is no missing data at this point.

	Number of patients enrolled in the cohort	Number (%) of patients With available data*	Number(%) of patients with actual missing data at the observation point
At least 2 months since enrollment (enrolled before September 2021)	968	789 (81.5)	168 (17.4)
At least 4 months since enrollment (enrolled before July 2021)	924	711 (76.9)	167 (18.1)
At least 6 months since enrollment (enrolled before May 2021)	766	551(71.9)	139 (18.1)
At least 8 months since enrollment (enrolled before March 2021)	576	329 (57.1)	74 (12.8)
At least 10 months since enrollment (enrolled before January 2021)	250	120 (48)	0

5) The presentation of the data is very superficial. Nothing is known about the clinical condition during the acute phase of infection.

Answer: We recognise this limitation. In this study, we focused on the prospective collection of patient reported outcomes during follow-up. Because some patients enrolled in the cohort more than six months after symptom onset, we made the methodological choice not to retrospectively ask for symptoms experienced during the acute phase of the infection to minimize memory bias.

We indicate this limitation in the discussion section of the paper.

Fourth, as patients could enrol in the cohort at any time point after their initial infection (with an interval between symptom onset and enrolment exceeding 300 days for 25% of participants), we chose to minimise memory bias by not asking retrospectively for symptoms experienced during the infection's acute phase. This limits our ability to describe how symptoms change immediately after the acute phase of the disease.

6) Since it was via an app, only a relatively young group that is tech-savvy. This possibly represents a bias.

Answer: We agree with the reviewer; our study using online questionnaires available on computer or smartphone included younger patients and more women. To improve the representativeness of descriptive and analytical results, we used statistical weighting methods

allowing to reconstruct a population-representative sample from non-representative samples obtained by surveys by weighting individual data using ancillary information (Kesse-Guyot, JMIR Public Health Surveill, 2016).

We used the recent data on the prevalence and the characteristics of patients with post COVID-19 disease, from the UK Office of National Statistics, as the ancillary information for the weighting procedure (Table 1 of the Dataset).

This is detailed in the manuscript and online methods.

To improve the representativeness of results, we weighted observations by calibration on margins, so that the weighted distribution of age (<24, 25-34, 35-49,5 0-69, and ≥ 70 years), gender, and hospitalisation during the acute phase of the disease (Yes/No) match the data from the UK Office of National Statistics (2 September 2021 data). Raw and weighted patients' characteristics are presented in Supplementary material 2. Hereafter, results presented always come from the weighted dataset. Raw results are available in the Supplementary materials.

The main manuscript now presents the results from the weighted analysis. Raw results for the evolution of symptoms and impact of the disease over time are presented in the supplementary material 4.

7) It is not a day-by-day assessment as the pat. were surveyed every 60 days via an app.

Answer: The reviewer is correct. We used the regular assessments of patients in the cohort to reconstruct the day-by-day evolution of symptoms. We believe that such approach, over a one-year time frame, would represent an important burden for patients and lead to poor retention in the follow-up. Our approach based on the reconstruction of the day-by-day prevalence of symptoms from evaluations every 60 days therefore represents a trade-off between feasibility for patients and precision in the assessments.

This specific point is also clarified in the main manuscript

We used multistate models taking both left and right censoring into account to estimate the longitudinal evolution of symptoms; each patient contributed to the data during his/her follow-up period. Because our cohort included patients enrolled at different times after their initial infection, we were able to reconstruct the prevalence of each symptom, day by day, in the study population

8) There was not a single face-to-face contact. It is a weakness that there are only online questionnaires without clinical assessment.

Answer: We respectfully disagree with the reviewer. In this study, we focused on symptoms and quality of life of patients, which are aspects that only patients can assess (Black, BMJ, 2013). Assessment of subjective phenomena, such as experiencing a symptom or not, may even be biased by clinician assessment who tend to underestimate patients' symptoms (Klopfenstein, Acta Anaesthesiol Scand, 2000). To this end, we used validated patient reported outcome measures, the validity and reliability of which were demonstrated in a previous study (Tran VT, Clin Infect Dis 2021).

Further, the literature tends to show that online questionnaires are less prone to social desirability bias than other methods of data collection, which makes them suitable for research on sensitive topics such as genitourinary symptoms or cognitive disorders (van Gelder, Am J Epidemiol 2010).

We have underlined these points in the discussion section of the paper

Strengths of this study lie in the prospective follow-up of a large population of patients with a confirmed COVID-19 infection and their regular assessment of their symptoms using validated patient-reported outcome measures, developed from patients' lived experience, with excellent reproducibility (intraclass correlation coefficient 0.83, 95% confidence interval 0.80 to 0.86)¹³. The use of online questionnaires reduced social desirability bias and may provide better estimates of the prevalence of sensitive symptoms such as genitourinary or cognitive disorders.

9) Finally, these observations do not suggest potential pathophysiological mechanisms underlying long COVID.

Answer: We agree and toned down the conclusion of the study accordingly.

Our results should be useful for researchers seeking the potential pathophysiological mechanisms underlying post COVID-19 disease.

Reviewer #2 (Remarks to the Author):

This is a longitudinal study of patients with persistent symptoms after COVID-19 in France. The authors have followed patients with confirmed SARS-CoV-2 infection every second month for up to one year, minimum 2 months. The study was online, meaning that participants were recruited by social media and patient associations. Recruitment required participants to having had symptoms for 3 weeks after diagnosis.

The recruitment procedure is the main weakness of the study. Having recruited 837 participants nationally, out of France's 6,8 million confirmed cases, makes risk of selection bias obvious. This is further confirmed by having achieved 80% women among the participants, while approx 50% of the infected are male. Statements in the abstract such as 88% having symptoms after 9 months can easily be interpreted as a prevalence, even though not specifically stated as such, and should be removed.

Answer: The ComPaRe long COVID cohort is a prevalent cohort of patients with post COVID-19 disease. It thus aims at representing patients with persistent symptoms and not patients who had had COVID-19 disease. It is also among the largest prospective cohorts of patients with persistent symptoms. In comparison to other research cohorts aimed at understanding the long term sequelae of COVID-19 reported in a recent review in Nature Medicine (Nalbandian, Nature Medicine 2021), only the COVIDOM (NCT04679584) and the NCT04573062 studies had larger sample sizes.

However, we agree that methods to recruit patients in our cohort may have led to selection bias.

To enhance the representativeness of descriptive and analytical results, we used statistical weighting methods enabling us to obtain a population-representative sample from non-representative survey samples by weighting individual data using ancillary information (Kesse-Guyot, JMIR Public Health Surveill, 2016). We used the recent data on the prevalence and the characteristics of patients with post COVID-19 disease, from the Office of National Statistics, as the ancillary information for the weighting procedure (Table 1 of the Dataset).

The main manuscript now presents the results from the weighted analysis. Raw results for the evolution of symptoms and impact of the disease over time are presented in the supplementary materials.

We also underline these points in the discussion

This study has several limitations. The recruitment of volunteers in the ComPaRe long COVID cohort may have selected patients who had overall more symptoms, and our recruitment strategy included a social media and general media campaign that may have selected younger and better educated patients. Second, our sample included a majority of women. Persistent symptoms after COVID-19 are more frequent among women; for example, the odds of persistent fatigue at one year among women compared with men was 1.43 [1.04-1.96]21. To minimise these issues, we used a weighted dataset with weights derived from the September 2021 data from the Office of National Statistics in the United Kingdom.

The interesting observations in the study are the observations of symptom changes over time. Again, a more rigid recruitment may have led to other changes in symptoms, but the observations are likely representing true longitudinal symptom patterns. The differences between younger and older participants is also of interest.

Specific comments:

1. Are all participants recruited 3 weeks after disease onset, or is it possible that some are recruited much later? This will affect the validity of the longitudinal data.

Answer: We apologise for the confusion. Patients could enrol in the cohort at any time after their initial infection. In fact, we leveraged the heterogeneity of disease duration to reconstruct the day-by-day prevalence of each symptom of post COVID-19 disease by using multi state models with both left and right censoring. In this study, each participant thus contributed to the data for the period of their follow-up. This has been clarified in the paper:

We used multistate models taking both left and right censoring into account to estimate the longitudinal evolution of symptoms; each patient contributed to the data during his/her follow-up period. Because our cohort included patients enrolled at different times after their initial infection, we were able to reconstruct the prevalence of each symptom, day by day, in the study population

We respectfully disagree with the reviewer that this affects the validity of the longitudinal data as we specifically used methods to account for heterogeneity of times after symptom onset. Further, our study was prospective, asking patients about their current symptoms and thus limiting memory bias. We used validated patient reported outcome measurements (Long COVID ST and IT, Tran VT, Clin Inf Dis 2021) which showed excellent content validity and reliability. Finally, we had only a limited number of patients lost to follow-up, limiting attrition bias.

2. What was the reasoning behind requiring 3 weeks duration of symptoms?

Answer: In the original paper, we defined post COVID disease as the persistence of symptoms 3 weeks after symptom onset. This choice was made in October 2020, when the cohort was planned and based on data available at that time (Tenforde, Morbidity Mortality Weekly Report, 2020). In the revised paper, we have reanalysed our data in light of the recent definition of post COVID-19 disease issued by the WHO on October 6, 2021, and restricted our analyses to patients with a confirmed infection and presenting at least one symptom for at least 2 months (60 days).

3. 80% of participants are women, and women have more symptoms of fatigue in general. Was any calculations performed on gender differences?

Answer: We thank the reviewer for the suggestion. Multiple studies have highlighted that men and women have a different risk of presenting persistent symptoms 1 year after an acute COVID-19 infection (Huang et al. Lancet 2021; Nehme M. Ann Intern Med 2021). Yet, to our knowledge, no study explored the differences in symptoms reported between men and women who already have persistent symptoms.

To further explore the gender difference suggested by the reviewer, we now report several exploratory subgroup analyses. We recommend that these results be interpreted with caution, as they are post-hoc and on small samples. Observed differences may thus be either due to a ‘real’ different evolution of symptoms between sexes, for example due to genetic factors (Takahashi, Nature 2020) and/or to different causes underlying the persistent symptoms.

We discuss the differences observed

We found some differences in the evolution of symptoms between men and women; and in age groups. We hypothesise that differences may be due either to different underlying causes of the persistent symptoms or to specific factors directly affecting symptoms. For example, several studies have highlighted differences between the sexes in immune responses (in terms of levels of innate immune cytokines such as IL-8 and IL-18, induction of non-classical monocytes, and T cell activation)¹⁷ and in response to a traumatic event.

4. Figure 1 shows cumulative remission of symptoms. If a substantial portion of participants were recruited much later than 3 weeks, there is a risk of over-representation of long-haulers. I would like to see an analysis of early and late recruited participants, if not all were recruited at 3 weeks (see my point 1 above).

Answer: The reviewer is correct. We apologize for the mistake and are grateful for having noticed it. As we used a prevalent cohort design, we only included in this study patients who had symptoms at baseline and our data were therefore left truncated. This mistake only affects the survival analysis as the description of symptoms used a multistate model that already accounted for left and right censoring.

We now account for this in the analyses. We have updated the online methods and corrected the results.

5. Figures 2 and suppl 5 are interesting. How much is the data affected by gender, and again, by recruitment time. Particularly the upwards trend from 6 to 9 months in fig 2B needs to be explained by more analysis regarding recruitment time after infection. Are there age and gender differences in the trends in this panel?

Answer: We analysed the evolution of the impact of the disease over time across groups defined by sex and age. We highlighted a similar trend in all subgroups. Differences involved mainly the intensity of the impact. For example, from 180 to 360 days about 60% of patients ≤ 40 years old reported that an important impact of their professional activities, whereas only 40% patients >40 years old reported such problem.

Regarding the trend from 6 to 12 month, it was more marked among younger patients, for whom post COVID-19 was more often their first contact with a chronic disease.

We discuss these points in the discussion.

Our results demonstrate the substantial impact of post COVID-19 disease on patients' lives. Specifically, the course of patients' perceptions of the impact of the disease changed similarly over time across all subgroups, with an aggravation six months after onset. We hypothesise that this corresponds to patients' realisation that this novel and previously unknown disease is chronic rather than acute. This trend seemed more pronounced among younger patients, for whom post COVID-19 is more likely to be both their first contact with a chronic disease and a biographical disruption of their lives.

6. Were any calculations made on the effect of comorbidity? I would like to see more statistical calculation on how symptom scores and duration were affected by comorbidity, age, gender, and other factors recorded as background variables.

Answer: We have also performed analyses subgroups of symptoms by comorbidity and provide results in the appendices.

As mentioned before, we believe that these analyses are mainly exploratory.

Reviewer #3 (Remarks to the Author):

Reviewer: Shamil Haroon

General comments

This paper describes a prospective cohort study on individuals with Long COVID who were SARS CoV-2 RT-PCR positive or had indicative serology. The study includes important and novel findings relevant to understand the natural history of Long COVID. The strengths of the study include the wide range of symptoms captured, the longitudinal data capture, as well as the inclusion of data on the impact of symptoms on quality of life.

A key limitation is the source population that was recruited by social media and not randomly sampled from a pre-specified sampling frame.

Answer: We recognise that recruitment may have led to selection bias. Yet, we want to emphasise that recruitment did not involve only social media. It also encompassed a traditional media (radio, newspapers) campaign and a call to participation on the French national contact tracing application. Multimodal recruitment methods in e-cohorts combining television/radio broadcasts have been shown to enable the recruitment of a diverse population in terms of socio-economic status, geographic distribution and literacy, who may not participate in more traditional research projects (Kesse Guyot, JMIR, 2013). Furthermore, a random sample would not preclude issues of representativeness. Even in a random sample, only patients who accept to participate are included and this population is no more a random sample.

To enhance the representativeness of descriptive or analytical results from our sample, we have used statistical weighting methods enabling us to obtain a population-representative sample from non-representative survey samples by weighting individual data using ancillary information (Kesse-Guyot, JMIR Public Health Surveill, 2016). We used the recent data on the prevalence and the characteristics of patients with post COVID-19 disease, from the UK Office of National Statistics, as the ancillary information for the weighting procedure (Table 1 of the Dataset).

There is also no control population to compare the difference in symptoms, making it difficult to infer the extent to which the pattern of symptoms observed can be attributed

to Long COVID as opposed to other factors such as comorbidities or social and environmental factors affecting the study population.

Answer: We agree that multiple factors may have influenced the evolution of symptoms and/or impact over time and that this is a limitation of our study.

A number of key reporting items were missing from the manuscript and it was sometimes difficult to assess because much of the key material is in the supplementary file. I would recommend writing the report in accordance with the STROBE statement for cohort studies as far as possible, although I appreciate that there are limits set by the journal format that may not allow this to be wholly possible within the main text.

Answer: In line with papers submitted to Nature Communications, methods were detailed in a specific document called “Online Methods”. We understand the difficulty for reviewers and readers to navigate through multiple documents and we integrated many of the elements presented in the supplementary materials in the revised version of the paper to clarify how results were obtained.

A STROBE Checklist is now also appended to the manuscript.

Specific comments

Abstract

- **The abstract does provide sufficient details on the study population. It should clarify if these are hospitalised or non-hospitalised patients, which setting they were recruited from (population-based, primary care, secondary care, critical care), whether they were adults or children, etc.**
- **The abstract should include a brief description of the Long COVID Symptom Tool. It is unclear whether it is a paper-based questionnaire, an online questionnaire or an app-based platform.**
- **The abstract reports the change in prevalence of symptoms over time but does not consistently state over what time period.**

Answer: We modified the abstract in accordance with the reviewer’s comments, while keeping the format and word limit (150 words) of Nature Communications.

About 10% of people infected by severe acute respiratory syndrome coronavirus 2 experience post COVID-19 disease. We analysed data from 968 adult patients (5350 person-months) with a confirmed infection enrolled in the ComPaRe long COVID cohort, a disease prevalent prospective e-cohort of such patients in France. Day-by-day prevalence of post COVID-19 symptoms was determined from patients’ responses to the Long COVID Symptom Tool, an

online validated self-reported questionnaire assessing 53 post COVID-19 disease symptoms. One year after symptom onset, 84.9% patients still reported their persistence, with a progressively lower prevalence of 27/53 symptoms (e.g., loss of taste/smell); 18/53 symptoms (e.g., dyspnoea) were stable, while the prevalence of 8/53 symptoms (e.g., paraesthesia) had increased. The disease impact on patients' lives began increasing 6 months after onset, as patients realized they had a chronic disease. Our results should be useful for researchers seeking the potential pathophysiological mechanisms underlying post COVID-19 disease.

Background

- **The first paragraph of the background section mentions that long COVID is the persistence of symptoms after initial SARS CoV-2 symptoms. I think this should be reworded to persistence of symptoms after initial SARS CoV-2 infection.**

Answer: This was a mistake and has now been corrected.

- **Note that the UK study referred to uses the term “post-covid-19 syndrome” rather than “post-acute COVID” to denote symptoms lasting beyond 12 weeks.**

Answer: We have replaced all occurrences of long COVID by “post COVID-19 disease”, as named by the WHO in a recent consensus document (A clinical case definition of post COVID-19 condition by a Delphi consensus, WHO, October 2021).

- **The background section mentions that only a handful of studies have investigated the longitudinal evolution of symptoms with Long COVID and that they are either small, retrospective or limited to single centres and cites two papers. The study by Davis et al that has been cited actually has quite a large sample size (n=3762) from 56 countries. Other relevant studies include Blomberg et al 2021 (Long COVID in a prospective cohort of home-isolated patients), and Sudre et al 2021 (Attributes and predictors of Long COVID).**

Answer: The different studies presented all have strengths and limitations. For example, the study from Davis et al. used a cross sectional evaluation and asked patients to recall the symptoms over a 6-month period, with a high risk of recall bias.

Other studies, such as the one from Blomberg (Nat Med 2021) or Sudre (Nat Med 2021) recruited patients at the acute phase of the infection and prospectively followed-them. Therefore, the number of patients who experienced post COVID-19 disease was smaller in their studies than in ours (189 patients with symptoms lasting >6 months in the study by Blomberg et al. and 95 patients with symptoms > 12 weeks in the study from Sudre et al.). Differences in results were also related to the measure of symptom persistence as none of the

aforementioned studies used validated measurement tools with known reliability to assess the disease.

In all, we present here the largest prospective study of the evolution of post COVID-19 disease over time with 968 patients followed over 5350 patient months with regular assessments of validated patient-reported outcomes.

Results

• It would be helpful to briefly mention whether the patients were recruited from a population-based cohort, from primary care, or secondary care, and the method of recruitment.

Answer: To summarise, the ComPaRe Long COVID cohort is an ongoing nationwide e-cohort of patients with post COVID-19 disease in France nested in the ComPaRe research programme (www.compare.aphp.fr), an umbrella e-cohort of patients with chronic conditions. Participants were informed of the existence of the research through 1) a social and general media campaign (mainly radio and newspapers), 2) calls for participation from partner patient associations and on the official French contact tracing app “TousAntiCOVID” (which is used by approximately 10.6 million persons in France), and 3) a "snowball" sampling method where participants would be encouraged to invite people who had a COVID-19 infection and persisting symptoms to enrol. This strategy avoided the potential pitfalls of relying on clinicians’ referral for recruitment (e.g., their lack of time and resources to cope with the additional workload of research duties) (Tran VT, J Clin Epidemiol 2020).

Participants interested in participating in ComPaRe can enrol themselves on the online website of the cohort (<https://compare.aphp.fr>). After completing initial demographic and clinical information, patients receive invitations by e-mail to answer online questionnaires related to their diseases and/or treatment, as well as prompts to participate in research projects nested in ComPaRe.

We have expanded the online methods to underline these details.

• When did study recruitment commence?

Answer: The ComPaRe long COVID cohort began in December 2020. This information was reported in the online methods and is now also presented in the main text.

The ComPaRe Long COVID cohort is an ongoing nationwide e-cohort of patients with post COVID-19 disease in France, nested in the ComPaRe research programme

(www.compare.aphp.fr), an umbrella e-cohort of patients with chronic conditions¹⁰. The cohort started in December 2020, and recruitment is ongoing.

• **Point 2 of the eligibility criteria states that participants should have had symptoms persisting three weeks after onset but doesn't clarify what is meant by "onset". Is this onset of symptoms or test result?**

Answer: We apologise for the imprecision. Onset referred to the date of symptom onset. We have clarified this throughout.

• **Further to the above point, what symptoms were included as part of the eligibility criteria and how were these symptoms selected?**

Answer: Eligibility was assessed as the presence of at least one symptom among the 53 assessed by the long COVID ST. The long COVID ST was developed from the experiences of 500 patients with long COVID (Tran VT, Clin Infect Dis 2021) and showed excellent reliability. This list also fully overlaps with those from the "Clinical Platform Case Report Form (CRF) for Post COVID condition (Post COVID-19 CRF)" of the WHO.

This was highlighted in the Online Methods.

In this study, we analysed the data from patients 1) reporting a laboratory-confirmed COVID-19 infection, with a positive test result for SARS-CoV2 by PCR swab and/or a serological assay; 2) reporting at least one symptom persisting two months after symptom onset, in a validated list of 53 symptoms³; and 3) who enrolled in the cohort by August 1, 2021, so that they would have at least 2 months of follow-up. The two-month interval was chosen according to the WHO's definition of post COVID-19 disease.

Those reporting that they still had persisting symptoms completed the long COVID symptom tool (ST) and impact tool (IT), a pair of validated patient-reported instruments assessing respectively 53 symptoms and 6 dimensions of patients' lives that can be affected by the disease³. Those reporting that they no longer had any symptoms were asked to specify the date when they first noticed the absence of symptoms.

Patients received reminders every 15 days encouraging them to complete the online questionnaires. Of note, the 53 symptoms of the long COVID ST fully overlap with those from the WHO's "Clinical Platform Case Report Form (CRF) for Post COVID condition (Post COVID-19 CRF)".

• **Did any participants die during follow-up? How was participant death dealt with in the analysis? Was follow-up censored at this point and how was this handled in the analysis?**

Answer: In ComPaRe, the vital status of participants is regularly assessed by linkage with the National database for deaths (INSEE registre des personnes décédées) (last update September 2021).

As of October 28, 2021, no patient enrolled in the ComPaRe long COVID cohort had died. This is not surprising because most patients enrolled had had a mild disease and only few patients were hospitalised during their acute COVID-19.

We have added these details in the online methods.

The vital status of ComPaRe participants is assessed regularly by linkage with the French National Death Register (INSEE, fichiers des personnes décédées). As of October 28, 2021, no patient enrolled in the cohort had died.

• What was the level of data completeness and how was missing data handled in the analyses? How complete was follow-up?

Answer: Available data and patients lost to follow-up are presented in the Supplementary material 3. In all, less than 20% of patients were lost to follow-up. Missing information was handled by right censoring in all analyses.

	Number of patients enrolled in the cohort	Number (%) of patients With available data*	Number(%) of patients with actual missing data at the observation point
At least 2 months since enrollment (enrolled before September 2021)	968	789 (81.5)	168 (17.4)
At least 4 months since enrollment (enrolled before July 2021)	924	711 (76.9)	167 (18.1)
At least 6 months since enrollment (enrolled before May 2021)	766	551(71.9)	139 (18.1)
At least 8 months since enrollment (enrolled before March 2021)	576	329 (57.1)	74 (12.8)
At least 10 months since enrollment (enrolled before January 2021)	250	120 (48)	0

• How was remission of symptoms classified? Was a single report of all symptoms resolving at any point during follow-up classed as remission or did the symptoms need to have resolved for a particular length of time?

Answer: As reported in the online methods, at each observation point (every 60 days), patients were first asked if they still had symptoms related to COVID-19. Those reporting that they still had persisting symptoms completed the long COVID symptom tool (ST) and impact tool (IT). Those reporting that they no longer had any symptoms are asked to specify the date when they first noticed the absence of symptoms. A single report of all symptoms resolving at any point during follow-up was enough to be classified as remission. This is underlined in the methods

Participants were followed-up every 60 days with online questionnaires. They received an e-mail invitation prompting them to connect on the ComPaRe (<https://compare.aphp.fr>) secure internet platform, which is accessible by computer or smartphone. At each observation point, patients were first asked if they still had symptoms related to COVID-19. Those reporting that

they still had persisting symptoms completed the long COVID symptom tool (ST) and impact tool (IT), a pair of validated patient-reported instruments assessing respectively 53 symptoms and 6 dimensions of patients' lives that can be affected by the disease³. Those reporting that they no longer had any symptoms were asked to specify the date when they first noticed the absence of symptoms.

And

First, the Kaplan-Meier method was used to estimate the cumulative probability of remission of all post COVID-19 symptoms. The date of symptom remission was defined as the first date when patients no longer reported any symptoms. A single report of all symptoms resolving at any point during follow-up was enough to be classified as remission. The time at risk started at disease onset and ended on October 10, 2021. Data were left truncated at the time of patients' enrolment in the cohort. Follow-up data were censored at the participants' latest observation point.

- **The probability of symptom persistence is reported at 6 and 9 months as 95.6% and 88.7%. Is this referring to those with a relapse of symptoms?**

Answer: The Kaplan Meier curve shows the time to first remission of symptoms and includes patients who subsequently relapsed.

- **Were subgroup analyses performed to assess whether the observed trends varied by specific characteristics such as age, sex, or hospitalisation status? The results seem to suggest this was done by age but there is no specific mention of this in the methods.**

Answer: We initially performed a subgroup analysis by age. As asked by the reviewer 2, we added post-hoc analyses in groups defined by sex and comorbidities. We did not perform a subgroup based on hospitalization status because of the limited number of patients hospitalized in our data. All subgroups analyses performed are now clearly indicated in the online methods. Results are presented in several new supplementary materials.

We performed several post-hoc subgroup analyses of the evolution of symptoms and impact on patients' lives by age (≤ 40 and > 40 years), sex and presence of at least one comorbidity (yes/no).

Figures

- **There appears to be an error in the y-axis title for figure 1. It indicates % of patients with remission of symptoms. However, this is actually the proportion rather than the percentage.**

Answer: We have corrected the error and thank the reviewer for pointing it out.

- **The opposite is true for the y-axis title for figure 2a. This states proportion of patients where it is the percentage.**

Answer: We have corrected the error.

- **The similarity of some of the colours in figure 2b make it difficult to interpret.**

Answer: We have changed the colours in Figure 2b.

Discussion

- **The authors suggest that the timeframes in NICE definitions of Long COVID should be revised. However, the rationale for this argument could be more clearly explained.**

Answer: We apologise for the imprecision. In our results, we see that most symptoms tend to decrease after the acute infection before plateauing. For example, the prevalence of coughing drops from 50% of participants to 20% within 4 months and then stays stable at this level. This is similar for loss of smell with a decrease from 50% to 20% within 8 months. These time frames are longer than what is usually considered in guidelines.

We have completely rewritten the paragraph.

The course of symptoms over time highlighted three distinct patterns that offer insight into the aetiologies and mechanisms underlying this disease. First, we observed a decrease in prevalence over time for symptoms such as loss of taste or smell, coughing, or diarrhoea. For example, the prevalence of coughing decreased from 50% to 20% of participants within the first 6 months after symptom onset before reaching a plateau. This was similar for loss of smell, with a plateau reached after 8 months. This evolution, indicating recovery from the acute phase, is slower than expected; indeed, most guidelines still consider a cut-off of 12 weeks to distinguish ongoing symptomatic COVID-19 (i.e., signs and symptoms of COVID-19 from 4 to 12 weeks) from post-COVID-19 syndrome

- **The explanation for a rise in prevalence of certain symptoms such as memory problems was also unclear. This could be phrased more clearly.**

Answer: We have reformulated the sentence.

For other symptoms, we showed that their prevalence increased over time. For example, the prevalence of hair loss increased over time with 8% and 15% of participants reporting it at 2 months and one year after onset respectively. Late symptom appearance, especially alopecia, has been reported in other studies and should be further investigated.

- **It would be helpful to guideline developers to include a clear statement on the implications of the findings for the management of Long COVID going forward.**

Answer: We believe that the most important message for clinicians caring for patients with long COVID is that the evolution of symptoms is a continuous and slow process.

In conclusion, our study shows that most patients with post COVID-19 disease have symptoms evolving in different patterns but persisting through one year. Recovery from the acute infection is a slow process, and the prevalence for most symptoms decreased over time before plateauing six to eight months after onset. Our results should be useful for researchers seeking the potential pathophysiological mechanisms underlying post COVID-19 disease and should also help physicians to inform their patients about the potential course of this disease.

Other

- **The sections on acknowledgements and contributions do not appear to have been completed.**

Answer: These sections were removed because of the blinded nature of peer review from Nature

REVIEWER COMMENTS

Reviewer #1 (Remarks to the Author):

The requirements are met and answered well and critically. I would therefore accept the work.

Reviewer #2 (Remarks to the Author):

The manuscript in its revised form is improved and my queries are clarified.

Since I as a reviewer misinterpreted the phrasing in the abstract regarding persisting symptoms among symptomatic patients at 2 months, I strongly recommend being more precise in the abstract, to state: Among patients symptomatic after 2 months, 85% still reported symptoms one year after their symptom onset.

In general, when asking 53 questions, the likelihood of having one or more symptoms at any time point is expected to be high even in a normal population. This needs to be stated as a weakness of the study.

Reviewer #3 (Remarks to the Author):

Reviewer: Shamil Haroon

General comments

I would like to thank the authors for revising the manuscript and taking on board many of the comments made by the reviewers and responding to them in detail. The revised manuscript is a significant improvement on the original manuscript and includes several additional analyses and analytical methods that enhance the study, such as the use of weighted adjustments using prevalence estimated from a large national COVID-19 survey, and the inclusion of several subgroup analyses. The methods have generally been clarified and the overall manuscript has been improved. I have only a few minor comments to add.

Specific comments

1. The abstract states that the disease impact on patients' lives began increasing 6 months after onset, as patients realized they had a chronic disease. This statement is unclear and suggests that the impact increased because patients considered that they had developed a chronic disease. I don't think the study has elicited the reason for why the disease impact increased 6 months from onset. Rather this is speculated in the results but not based on any qualitative data collection. I would therefore suggest removing this statement from the abstract.

2. The abstract states that the findings of the study will be useful for understanding the potential pathophysiological mechanisms of Long COVID. However, the study does not address pathophysiology

but rather the findings will be useful for understanding the natural history of this syndrome. This also applies to the conclusion in the discussion.

3. A limitation of the study is that there was no control group. It is therefore impossible to ascertain whether the observed symptoms are different from or in addition to what might be due to comorbidities, ageing, or social effects of living through the pandemic. A more robust study design would include a comparable control group. This limitation has not been acknowledged in the discussion and ideally should be.

4. How were the four symptoms highlighted in figure 2A selected? They are not all the most prevalent symptoms, so it is unclear why these four symptoms were specifically highlighted.

5. The following text has now been added to the manuscript: "To improve the representativeness of results, we weighted observations by calibration on margins, so that the weighted distribution of age (<24, 25-34, 35-49, 50-69, and ≥70 years), gender, and hospitalisation during the acute phase of the disease (Yes/No) match the data from the UK Office of National Statistics (2 September 2021 data)." Please clarify that the weights have been derived from data from the UK Office of National Statistics Covid Infection Survey.

Please find below a point-by-point answer to all reviewers' comments for our manuscript titled "*Course of post COVID-19 disease symptoms over time in the ComPaRe long COVID prospective e-cohort*" (NCOMMS-21-29487A-Z).

Reviewer #1 (Remarks to the Author):

The requirements are met and answered well and critically. I would therefore accept the work.

We thank the reviewer for his previous comments, which greatly helped us improve the manuscript

Reviewer #2 (Remarks to the Author):

The manuscript in its revised form is improved and my queries are clarified.

Since I as a reviewer misinterpreted the phrasing in the abstract regarding persisting symptoms among symptomatic patients at 2 months, I strongly recommend being more precise in the abstract, to state: Among patients symptomatic after 2 months, 85% still reported symptoms one year after their symptom onset.

We agree that the sentence was misleading and modified it to match the reviewer's suggestion.

About 10% of people infected by severe acute respiratory syndrome coronavirus 2 experience post COVID-19 disease. We analysed data from 968 adult patients (5350 person-months) with a confirmed infection enrolled in the ComPaRe long COVID cohort, a disease prevalent prospective e-cohort of such patients in France. Day-by-day prevalence of post COVID-19 symptoms was determined from patients' responses to the Long COVID Symptom Tool, an online validated self-reported questionnaire assessing 53 post COVID-19 disease symptoms. Among patients symptomatic after 2 months, 85% still reported symptoms one year after their symptom onset. Evolution of symptoms showed a decreasing prevalence over time for 27/53 symptoms (e.g., loss of taste/smell); a stable prevalence over time for 18/53 symptoms (e.g., dyspnoea), and an increasing prevalence over time for 8/53 symptoms (e.g., paraesthesia). The disease impact on patients' lives began increasing 6 months after onset, as patients realized they had a chronic disease. Our results should be useful for researchers seeking the potential pathophysiological mechanisms underlying post COVID-19 disease.

In general, when asking 53 questions, the likelihood of having one or more symptoms at any time point is expected to be high even in a normal population. This needs to be stated as a weakness of the study.

We agree with the reviewer and now clearly state this as a limitation of the study.

Fifth, disease remission was defined as having no symptoms among 53. As the likelihood of having at least one symptom, at any time point, is expected to be high even in the general population, our analyses may overestimate the number of patients who still report symptoms one year after their symptom onset

Reviewer #3 (Remarks to the Author):

Reviewer: Shamil Haroon

General comments

I would like to thank the authors for revising the manuscript and taking on board many of the comments made by the reviewers and responding to them in detail. The revised manuscript is a significant improvement on the original manuscript and includes several additional analyses and analytical methods that enhance the study, such as the use of weighted adjustments using prevalence estimated from a large national COVID-19 survey, and the inclusion of several subgroup analyses. The methods have generally been clarified and the overall manuscript has been improved. I have only a few minor comments to add.

We thank the reviewer for the comment

Specific comments

1. The abstract states that the disease impact on patients' lives began increasing 6 months after onset, as patients realized they had a chronic disease. This statement is unclear and suggests that the impact increased because patients considered that they had developed a chronic disease. I don't think the study has elicited the reason for why the disease impact increased 6 months from onset. Rather this is speculated in the results but not based on any qualitative data collection. I would therefore suggest removing this statement from the abstract.

We agree that the statement was speculative and removed the sentence from the abstract.

2. The abstract states that the findings of the study will be useful for understanding the potential pathophysiological mechanisms of Long COVID. However, the study does not

address pathophysiology but rather the findings will be useful for understanding the natural history of this syndrome. This also applies to the conclusion in the discussion.

We modified the abstract according to the reviewer's comments.

Abstract:

About 10% of people infected by severe acute respiratory syndrome coronavirus 2 experience post COVID-19 disease. We analysed data from 968 adult patients (5350 person-months) with a confirmed infection enrolled in the ComPaRe long COVID cohort, a disease prevalent prospective e-cohort of such patients in France. Day-by-day prevalence of post COVID-19 symptoms was determined from patients' responses to the Long COVID Symptom Tool, an online validated self-reported questionnaire assessing 53 post COVID-19 disease symptoms. Among patients symptomatic after 2 months, 85% still reported symptoms one year after their symptom onset. Evolution of symptoms showed a decreasing prevalence over time for 27/53 symptoms (e.g., loss of taste/smell); a stable prevalence over time for 18/53 symptoms (e.g., dyspnoea), and an increasing prevalence over time for 8/53 symptoms (e.g., paraesthesia). The disease impact on patients' lives began increasing 6 months after onset. Our results are of importance to understand the natural history of post COVID-19 disease.

Conclusion

Our results are of importance to understand the natural history of this disease, and should help physicians to inform their patients about the potential course of this disease.

3. A limitation of the study is that there was no control group. It is therefore impossible to ascertain whether the observed symptoms are different from or in addition to what might be due to comorbidities, ageing, or social effects of living through the pandemic. A more robust study design would include a comparable control group. This limitation has not been acknowledged in the discussion and ideally should be.

We agree that, without a control group, we cannot ascertain that the observed symptoms are caused by post COVID-19 disease and not intercurrent illnesses, comorbidities, ageing, or social effects of living through the pandemic; especially as many symptoms are non-specific and prevalent in the general population.

This limitation is highlighted in the discussion.

Sixth, this study did not use a control group. Because several symptoms of post COVID-19 disease are non-specific, we cannot ascertain that the observed symptoms are different from or in addition to what might be due to intercurrent illnesses, comorbidities, ageing, or social effects of living through the pandemic.

4. How were the four symptoms highlighted in figure 2A selected? They are not all the most prevalent symptoms, so it is unclear why these four symptoms were specifically highlighted.

The four symptoms highlighted in figure 2A were chosen to represent different important aspects of post COVID-19 disease.

- Fatigue is the most commonly reported symptom among patients with post COVID-19 disease.
- Loss / change of taste embodies “specific” symptoms of long COVID alongside loss / change of smell. We chose to highlight the change of taste because it was more prevalent 60 days after the onset of symptoms.
- Cough, thoracic pain and dyspnoea can embody the potential sequelae of the acute disease. Again, we chose cough because it was the most prevalent at 60 days.
- Finally, among the 8 symptoms with an increasing prevalence, we chose to highlight paraesthesia over neck / back and low back pain because it seemed, for us, more specific of post COVID-19 disease.

5. The following text has now been added to the manuscript: “To improve the representativeness of results, we weighted observations by calibration on margins, so that the weighted distribution of age (<24, 25-34, 35-49,5 0-69, and ≥70 years), gender, and hospitalisation during the acute phase of the disease (Yes/No) match the data from the UK Office of National Statistics (2 September 2021 data).” Please clarify that the weights have been derived from data from the UK Office of National Statistics Covid Infection Survey.

We modified the text accordingly

To improve the representativeness of results, we weighted observations by calibration on margins, so that the weighted distribution of age (<24, 25-34, 35-49,5 0-69, and ≥70 years), gender, and hospitalisation during the acute phase of the disease (Yes/No) match the data from the UK Office of National Statistics Covid Infection Survey (2 September 2021 data)

REVIEWERS' COMMENTS

Reviewer #3 (Remarks to the Author):

I would like to thank the authors for their further revision of the manuscript, which appropriately addressed all outstanding feedback from the reviewers.

Dr Shamil Haroon